# THE TUTOR-PUPIL AUGMENTATION: ENHANCING LEARNING AND INTERPRETABILITY VIA INPUT CORRECTIONS

**Darya Biparva**
Department of Electrical and
Computer Engineering
University of Minnesota
Minneapolis, MN 55455
`bipar001@umn.edu`

**Maarten Schoukens**
Department of Electrical Engineering
Eindhoven University of Technology
Eindhoven, Netherlands
`m.schoukens@tue.nl`

**Donatello Materassi**
Department of Electrical and
Computer Engineering
University of Minnesota
Minneapolis, MN 55455
`mater013@umn.edu`

## ABSTRACT

State-of-the-art machine learning models often incorporate prior knowledge or structural information about the task or data distribution. In some tasks, such knowledge may arise from first principles or emerge as simplified, learned functions that distill essential aspects of the data distribution. Model augmentation has emerged as a strategy to leverage this structured knowledge by coupling it with an auxiliary model to improve predictive performance, while preserving the interpretability offered by the simpler component. In this work, we present a new augmentation framework called the Tutor-Pupil scheme, which is designed to enhance both performance and interpretability. The Pupil is a fixed model, structurally designed for the core task, while the Tutor is a more flexible model trained to apply minimal input-level corrections to improve the Pupil's performance on the modified input. This strict separation of roles enables the Tutor not only to compensate for the Pupil's limitations but also to act as a diagnostic instrument. By examining the Tutor's targeted interventions, we can identify failure modes, detect regions where the Pupil struggles to generalize, and uncover residual patterns or higher-order structures in the data not captured by the original model.

## 1 INTRODUCTION

Recent advances in machine learning and artificial intelligence have led to the proliferation of many specialized model architectures, each designed to excel at particular classes of tasks. For example, convolutional neural networks (CNNs) (LeCun et al., 1989; Li et al., 2021; Anwar et al., 2018) have proven highly effective for visual data, leveraging spatial locality and translation invariance to classify and manipulate images. Transformers have revolutionized natural language processing by capturing long-range dependencies through attention mechanisms (Vaswani et al., 2017; Patwardhan et al., 2023). Recurrent neural networks (RNNs) (Rumelhart et al., 1985; Jordan, 1997) and their variants (Waqas & Humphries, 2024) continue to serve time-dependent modeling tasks, while more recent paradigms, such as liquid neural networks (Hasani et al., 2021) and diffusion models (Sohl-Dickstein et al., 2015; Croitoru et al., 2023), expand the landscape of architectures capable of dynamically adapting to new inputs or modeling generative processes. The common thread across these architectures is their ability to exploit known patterns and structural relations embedded in domain-specific data, which can be seen as a form of a priori information about the task (Hsieh et al., 2024; Aditya et al., 2019).

An alternative strategy for incorporating a priori knowledge into machine learning models is offered by Physics-informed neural networks (PINNs) (Raissi, 2018) and physics-guided neural networks (PGNNs) (Daw et al., 2022). Rather than relying solely on architectural design, PINNs and PGNNs inject domain knowledge by enforcing physical constraints, either explicitly through differential equations in the loss function, or implicitly through the use of physics-based features, relationships, or regularization (Karniadakis et al., 2021; Willard et al., 2022). These constraints guide the model toward physically plausible solutions even when data is relatively scarce or noisy. In both cases, whether through architecture or training objectives, success hinges on the model's ability to internalize and utilize structural information about the underlying phenomena (Von Rueden et al., 2021). However, while a carefully chosen architecture or the incorporation of correct first principles in a model can capture substantial structure present in the data, it does not necessarily capture all relevant patterns, especially those that are more subtle, or unanticipated (Krishnapriyan et al., 2021; Karpatne et al., 2024).

To further improve predictive performance and generalization, a recent line of research has explored model augmentation, where a primary model is enhanced through the addition of a secondary, auxiliary model (Ghosh et al., 2019). The interconnection between the two models can take various forms, including series, parallel, and feedback configurations, as well as more intricate composite structures (Sun et al., 2020; Götte & Timmermann, 2022; Groote et al., 2022; Shah et al., 2022; Hoekstra et al., 2024; Györök et al., 2025). The primary model is typically chosen to reflect the structural characteristics or prior knowledge suited to the task (for instance, a CNN for image data or a dynamical model for time-series), while the auxiliary model keeps a more flexible architecture in order to capture secondary patterns or unknown relationships in the data. Indeed, model augmentation has been extensively used in settings where the primary model is derived from scientific knowledge (for example, pyrolysis equilibrium model in (Jiang et al., 2024) and low resolution atmospheric global circulation model in (Arcomano et al., 2022)) allowing the auxiliary model to compensate for discrepancies between theoretical assumptions and observed data (Sharma & Liu, 2022).

A related approach is that of ensemble modeling, where multiple models of comparable capacity are trained and combined to improve robustness and accuracy. In contrast to model augmentation, which establishes a structured interaction between a primary model (often grounded in a priori knowledge) and an auxiliary model, ensembles aggregate diverse learners to mitigate variance and bias, albeit often at the expense of interpretability (Zhou, 2025; Freund, 1999).

The idea also bears resemblance to residual learning (He et al., 2016; Shafiq & Gu, 2022). In residual networks, such as ResNet, the architecture is designed to learn a residual mapping instead of the original transformation, primarily to ease optimization and improve gradient flow in deep models. These residual connections operate in the hidden feature space and serve an optimization purpose rather than a modeling one. By contrast, in model augmentation the primary model is typically fixed based on prior knowledge, and the auxiliary model learns a correction in the output space to compensate for discrepancies between theoretical assumptions and observed data.

In this work, we adopt the model augmentation paradigm but shift its focus toward a new goal: interpretability and explainability. Rather than viewing the auxiliary model exclusively as a corrective mechanism to boost accuracy, we treat it also as an interpretive tool that enhances transparency and insight. Towards this goal, we introduce a specific augmentation scheme, which we call the Tutor-Pupil scheme. In this scheme, the Pupil is the task-specialized model, chosen to match the structure of the application data, while the Tutor is a more flexible architecture that learns to model second-order discrepancies, that is, residual patterns not captured by the Pupil. Crucially, the Tutor is trained to make these corrections minimally by intervening only to the extent necessary to improve the Pupil's output.

This separation of roles offers a powerful framework for both improving performance and probing model behavior. Because the Tutor's interventions are designed to be as limited as possible, and since the Pupil remains unchanged during the training of the Tutor model, their structure becomes informative: they expose where and how the Pupil's reasoning fails or needs refinement. By analyzing the Tutor's corrections, we can identify specific failure modes, uncover generalization gaps, and explore regions of uncertainty in the input space. Moreover, when the Pupil is grounded in known scientific or domain-specific principles, such as in the case of first-principles models, the Tutor's corrective patterns may reveal higher-order structures or unmodeled effects. These insights can then be extracted and formalized using symbolic regression (Kronberger et al., 2024) or model-based explainable AI techniques (Ribeiro et al., 2016), effectively expanding the original model's conceptual

foundation. In this way, the Tutor does not merely enhance performance, but instead becomes a bridge between empirical behavior and theoretical understanding.

## 2 TUTOR AUGMENTATION OF AN INTERPRETABLE PUPIL

To concretely illustrate the Tutor-Pupil architecture, we begin with a simple yet representative toy example: a binary classification task in which the Pupil model is implemented as a decision tree. Decision trees are considered inherently interpretable, but often suffer from limited expressiveness, particularly when faced with complex or noisy decision boundaries. By augmenting this base classifier with a more flexible Tutor model, we aim to demonstrate how minimal input-level corrections can substantially improve classification performance while preserving the interpretability of the original decision process. Specifically, consider an elementary binary classification problem, as shown in Figure 1, in which features are denoted by $x_1$ and $x_2$ and data points with label $y = 1$ are represented by '+' marker while those with label $y = 0$ are represented by 'o' markers. The unknown true decision boundary is shown with a green line.

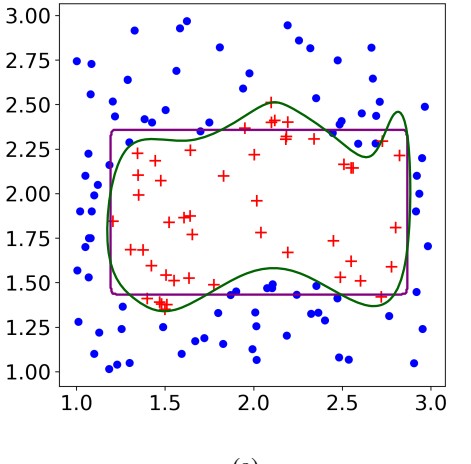 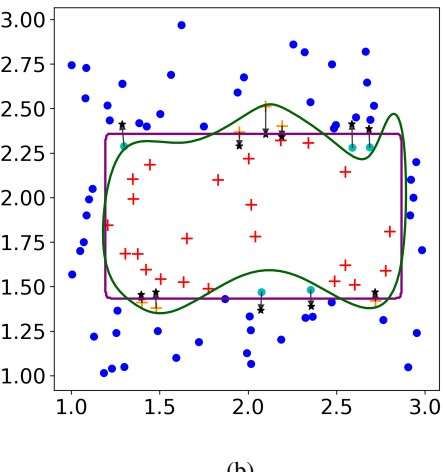

(a)                                   (b)

Figure 1: (a) represents the ground truth boundaries of a binary classification problem. The polygon represents the interpretable function $f$ that performs classification. Data represented in orange and light blue in (b) are data that the augmented model classified them differently than the interpretable model. The minimal change $\epsilon$ identified by the neural network is depicted with a black arrow, and the modified data point $x + \epsilon$ is depicted with a black star. To make the figure less crowded, a subset of data points are shown in this figure.

To perform this classification, we fit a shallow decision tree, which makes predictions based on thresholding values of input features, to the data. The decision boundaries induced by such a tree partition the feature space into polytopes within which the model's output remains constant. Assume that the decision tree maximizing classification accuracy on the training set has been selected and is represented by the function $\tilde{f}(x_1, x_2)$. The decision boundaries defined by $f$ are depicted in Figure 1(a), highlighted with a purple stroke.

The principal merit of this approach lies in its transparency: each parameter of the model admits a complete interpretation in terms of its effect on the decision boundary, rendering the model not just a predictive device but an intelligible explanation. Despite its simplicity, the decision tree succeeds in capturing a substantial portion of the data distribution. Yet, as Figure 1(a) shows, the model fails to account for certain subtleties in the distribution, particularly at its margins.

On the other hand, we could turn to neural networks or other more complex machine learning models that have a larger parameter space and achieve better accuracy. However, this would come at the cost of interpretability, as the decision boundary, though potentially more accurate, would be more opaque as the contribution of individual parameters becomes less clear. Thus, we are presented with

a trade-off between the accuracy afforded by complex models and the intelligibility characteristic of simpler ones.

It is precisely this trade-off that motivates the notion of model augmentation. The guiding observation is that simple, interpretable models, though limited in expressive capacity, often succeed in capturing meaningful low-order structure within the data. Rather than discarding such a model because of its limited accuracy, we may retain it as a primary model and assign to a more complex auxiliary model the task of accounting for the residual error that the primary model cannot account for.

Several augmentation schemes have been proposed. Among them, the parallel configuration, depicted in Figure 2(a), is the most prevalent. In this scheme, an auxiliary model is trained to minimally perturb

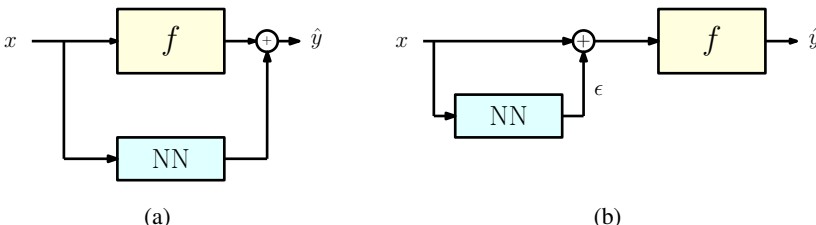

Figure 2: (a) Parallel augmentation: The auxiliary model is trained to directly correct the output of the primary model, improving performance but limiting interpretability—especially in regions where the auxiliary model dominates. (b) Tutor-Pupil augmentation: The auxiliary Tutor model applies minimal corrections in the input space, enabling the Pupil to make accurate predictions. This input-level intervention provides a more interpretable explanation of the primary model's failure modes and highlights how decision boundaries might need to shift.

the output of the primary model $f$, improving overall accuracy while preserving interpretability when possible. While this approach can indeed boost performance, it operates by modifying the output directly. As a result, in regions where the primary model $f$ performs well, the prediction is primarily determined by the interpretable function $f$, preserving its interpretability. However, in regions where $f$ performs poorly, the auxiliary model dominates the prediction. Since this model is typically opaque, interpretability is lost in precisely the regions where the model's reasoning is most in question. Even if the correction is kept small, this scheme only reveals how far the output of the interpretable model must be adjusted, offering little insight into why the model fails or what structure underlies the residuals.

In our view, a more informative alternative is provided by the Tutor-Pupil augmentation scheme shown in Figure 2(b), which we propose and study in this work. In this configuration, the auxiliary model, the Tutor, suggests a small perturbation $\epsilon$ to the input $x$, such that the Pupil (the primary, task-specialized model) produces the correct output when evaluated at $x + \epsilon$. Since the correction $\epsilon$ is applied in the input space, it lends itself more readily to human interpretation, offering an intuitive understanding of where the Pupil model's reasoning broke down and how it can be amended. In other words, rather than adjusting predictions post hoc, the Tutor helps reveal and compensate for the Pupil's limitations in a way that illuminates the decision process, especially in regions where the Pupil "struggles."

We can visualize and illustrate the behavior of this Tutor-Pupil augmentation scheme using our toy binary classification example. The Pupil model, a decision tree, assigns label $y = 1$ to all points that fall within the polytope shown in Figure 1(a). However, some points with the true label $y = 0$ lie inside this region and are therefore misclassified. The role of the Tutor, implemented here as a neural network, is to learn small perturbations $\epsilon$ to the input such that the modified input $x + \epsilon$ is correctly classified by the Pupil. For misclassified points with $y = 0$, this means nudging the input just outside the decision boundary to achieve $f(x + \epsilon) = 0$, while keeping $\epsilon$ minimal. Conversely, for points with $y = 1$ lying outside the region, the Tutor learns perturbations that shift them inward, correcting the classification to $f(x + \epsilon) = 1$.

This behavior is encouraged through the following loss function used to train the Tutor

$$\mathcal{L} = \underbrace{\mathcal{L}_{\mathrm{C}}(y, f(x + \epsilon))}_{\text{classification}} + \lambda \underbrace{\|\epsilon\|_2^2}_{\text{correction magnitude}}$$

where $\mathcal{L}_C(y, f(x + \epsilon))$ is the classification loss (e.g. binary cross entropy), and $\lambda$ is a regularization parameter that limits the correction magnitude. On average, this augmentation improves test accuracy by about $13\%$. Implementation details, along with the full set of performance metrics and statistical tests used to assess these changes, are provided in Appendix A.

The result of this training is shown in Figure 1(b) (to make the figure less crowded, a subset of data points is shown in this figure). Misclassified points ($f(x) \neq y$) are highlighted in light blue for $y = 0$ and orange for $y = 1$. Black arrows indicate the learned perturbation vectors $\epsilon$, and the resulting modified inputs $x + \epsilon$ are marked with black stars. Consider, for example, the data point $x = (2.05, 1.5)$ with the true label $y = 0$. Since $x$ lies within the Pupil's decision boundary, we have $f(x) = 1$. The Tutor proposes a correction $\epsilon = (-0.0082, 0.0960)$, which yields $x + \epsilon = (2.0418, 1.5960)$ which the Pupil now classifies correctly as $f(x + \epsilon) = 0$. This perturbation is largely along the $x_2$-axis, suggesting that the local data distribution deviates from the Pupil's axis-aligned decision boundary, bending inward along that axis. Such directional corrections offer insight into how the decision boundary should deform to better match the true data structure.

In this respect, we would like to emphasize that the Tutor-Pupil scheme enables a form of interpretability that is inherently more global than traditional, local explanation methods such as counterfactuals (Dai et al., 2022; Stepin et al., 2021). While counterfactual explanations typically focus on how small changes in individual features affect a single prediction, the Tutor is trained across the entire dataset to learn coherent patterns of correction that apply broadly across the input space. Rather than probing the sensitivity of the output to feature-level perturbations, the Tutor acts as a learned, input-level corrective mechanism that identifies systematic discrepancies between the Pupil's decisions and the true labels. This allows for multi-feature interventions that reflect higher-order structure in the data, offering insights into general failure modes of the Pupil model rather than isolated explanations tied to individual instances.

## 3 INTERPRETABLE AUGMENTATION OF FIRST PRINCIPLE FUNCTIONS

The Tutor-Pupil augmentation framework described in the previous section is not limited to data-driven interpretable models. It can also be applied to functions derived from first principles, such as those based on physical laws. These models, though principled and interpretable by design, often rely on idealized assumptions that inevitably limit their accuracy when confronted with complexities of real-world phenomena. To illustrate this broader applicability of Tutor-Pupil augmentation, consider the example of the behavior of an ideal gas as modeled from thermodynamic principles. The state of such a system is determined by three macroscopic variables: pressure ($P$), volume ($V$), and temperature ($T$) linked through the well-known equation of state:

$$P = \frac{nRT}{V},$$

where $R$ is the universal gas constant and $n$ is the number of moles of gas (Hill, 2013). This relation, theoretically derived from statistical mechanics, captures essential features of gaseous behavior under several key assumptions: particles are modeled as having infinitesimal radius, interact only via elastic collisions (ensuring energy conservation in accordance with the first law of thermodynamics) and follow Hamiltonian dynamics, which enables the application of Liouville's theorem and the $H$-theorem (Tolman, 1979; Callen, 2006), thereby implying the second law of thermodynamics by ensuring non-negative entropy changes under adiabatic transformations (Chandler, 1987; Thompson, 2015). Additionally, no interaction potential is assumed between particles beyond their instantaneous collisions (Tolman, 1979; Chandler, 1987). Such a relation serves as a remarkably effective approximation for many scenarios, yet, like all idealized models, its predictions may exhibit significant deviation from real-world behavior when its underlying assumptions no longer hold.

To explore this point concretely, we conducted a simulation of spherical particles with finite radius $r > 0$ confined within a two-dimensional box of volume $V$. All collisions, both between particles and with the container walls, were modeled as perfectly elastic. An illustration of the simulated system is provided in Figure 3(a). This setup was designed to offer a microscopic perspective on gas behavior, bridging particle-level dynamics with macroscopic quantities. Indeed, we defined the system's temperature $T$ as the average kinetic energy of the particles and computed pressure $P$ based on the average momentum exchanged between particles and the container walls (see the supplementary material for full details of the simulation procedure). With these definitions in place, we compared the macroscopic predictions of the ideal gas law to the behavior observed in our microscopic simulation. To visualize the results, we plotted isotherms (Lewis & Randall, 2020),

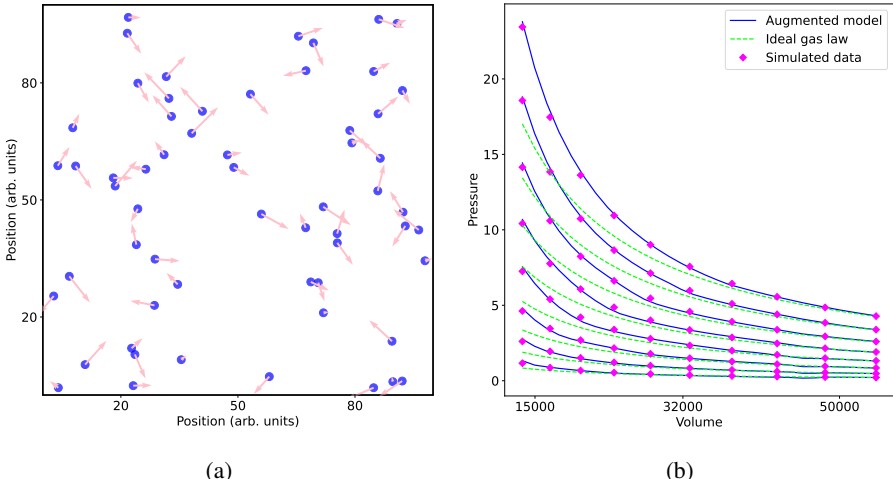

Figure 3: (a) shows a visualization of the simulation performed, where each particle is shown with a dot and the velocities are shown as the vectors. (b) shows the isotherms of the simulated data, shown in pink, ideal gas law, shown in green, and the augmented model shown in blue. We can see that ideal gas law underestimates the pressure of the gas in small container volumes.

curves of pressure versus volume at constant temperature, that are commonly used to characterize how a system responds to compression or expansion under thermal equilibrium. Figure 3(b) shows the resulting pressure-volume isotherms from the simulation alongside the theoretical curves predicted by the ideal gas law.

While the ideal gas law offers a reasonable approximation of pressure at large volumes, we observe a systematic underestimation of pressure as the volume decreases. Also, this discrepancy becomes more pronounced as the temperature increases. Rather than discarding this well established model, we decided to treat it as a Pupil and introduced a Tutor (in the form of a neural network) learning small data driven input corrections, slight shifts in volume and temperature, which we denote as $\epsilon = (\epsilon_V, \epsilon_T)$ to improve the model predictive performance. Specifically, the Tutor model was trained to minimize the following loss function

$$\mathcal{L} = \underbrace{\left(\frac{nR(T + \epsilon_T)}{V + \epsilon_V} - P\right)^2}_{\text{prediction loss}} + \lambda \underbrace{\|\epsilon\|_2^2}_{\text{correction magnitude}} .$$

The predicted pressure of the Tutor-Pupil model is given by $\hat{P} = \frac{nR(T+\epsilon_T)}{V+\epsilon_V}$ and is drawn in Figure 3(b), tracking almost perfectly the experimental pressure. Since the outputs of the Tutor belong to the input space, we can investigate the nature of the applied corrections, that are illustrated in Figure 4. The original temperature and volume are represented as red dots, while the change $\epsilon$ is shown as the purple vector. The corrected data points are marked in blue. The applied corrections become more significant at smaller volumes. Notably, the Tutor essentially prescribes, for small volumes, further reductions in effective volume to improve the accuracy of the Pupil model's predictions.

Recognizing this pattern, the physical rationale behind the correction becomes clearer: it addresses a key violated assumption of the ideal gas law, namely, that gas molecules have negligible size and can explore the full container volume without restriction. In reality, when particles have non-negligible radii, their centers cannot come closer to the walls than their own radius, effectively shrinking the accessible volume. This reduction becomes more pronounced in smaller containers (or with larger particles), leading to systematic deviations from the predictions of the ideal gas law.

In the scientific literature, an important refinement of the ideal gas law is given by the *van der Waals equation* (Klein, 1974), which introduces corrective terms to account for the finite size of particles and the intermolecular forces that become significant under high-density or low-temperature conditions. The van der Waals relation is typically written as:

$$P = \frac{nRT}{(V - b)} - \frac{a}{V^2}$$

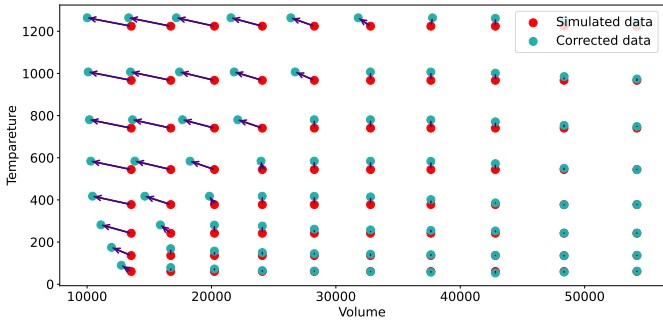

Figure 4: The plot shows the volume and temperature corrections $\epsilon_V$ and $\epsilon_T$ as learned by the Tutor. Larger corrections are observed at smaller volumes, where the assumptions of the ideal gas law break down due to finite particle size. The Tutor learns to systematically reduce the effective volume to account for excluded space.

where the parameters $a$ and $b$ are empirical constants that vary across different substances. The constant $b$ accounts for the finite volume occupied by the gas particles themselves, effectively reducing the space available for molecular motion, while $a$ captures the effect of intermolecular attractions, which tend to reduce the pressure exerted on the walls.

Interestingly, the data-driven Tutor-Pupil framework naturally leads to analogous insights. As shown in the simulation results, the Tutor learns to correct the predictions of the ideal gas law in regimes where its assumptions break down, especially at small volumes. These corrections manifest as volume reductions, reminiscent of the role played by the $b$ term in the van der Waals formulation.

Furthermore, if symbolic equation discovery methods (e.g., sparse regression or neural symbolic regression) were applied to the learned corrections provided by the Tutor, one could recover expressions structurally similar to the van der Waals equation. However, unlike the classical form where $a$ and $b$ are constant parameters, the data-driven approach could yield analytical expressions for the volume and temperature corrections $\epsilon_V$ and $\epsilon_T$, potentially as functions of $V$, $T$. This provides a richer, more adaptive model that maintains interpretability while being able to capture secondary principles to effectively discover models for non-ideal effects.

## 4    TUTOR AS AN EXPLANATORY TOOL FOR A NON-INTERPRETABLE PUPIL

In the previous sections, we illustrated how the Tutor-Pupil architecture can enhance both performance and interpretability when the Pupil is an inherently interpretable model, either by design or by virtue of its connection to first principles. In those settings, the Tutor acts as a corrective mechanism that not only improves accuracy but also reveals where and why the Pupil fails, thereby potentially offering additional insights into the underlying data or physical processes.

In this section, we turn our attention to a different scenario, where the Pupil model is not readily interpretable, either due to its internal complexity or the high dimensionality of its input space. In such cases, the Tutor's explanatory role shifts. Rather than providing insights on the data or the physical world, the Tutor becomes a diagnostic tool that helps us understand the inner workings of the Pupil itself. By learning to apply small, targeted input modifications that improve the Pupil's performance, the Tutor effectively highlights what features the Pupil is sensitive to and what information it requires to succeed. In this way, the Tutor acts as a probe into the Pupil's behavior, illuminating its implicit strategies, underlying mechanisms, and points of uncertainty.

To examine this setting in practice, we consider the task of handwritten digit classification using the MNIST dataset (LeCun et al., 1998). While this benchmark is well-studied and many modern models can achieve near-perfect accuracy on it, we intentionally select a logistic regression classifier as the Pupil model. This choice is motivated by several factors: first, logistic regression is a classical and well-understood method, which allows us to validate whether the Tutor's interventions align with known decision boundaries and model sensitivities. Second, despite its simplicity, logistic regression still operates in a high-dimensional input space (each $28 \times 28$ image yields 784 features), making it difficult to interpret directly in terms of raw parameters. This setup strikes a balance between

complexity and interpretability: it is complex enough to be opaque to human intuition when applied to image data, yet simple enough for us to assess whether the Tutor's learned corrections reflect the underlying structure of the Pupil's decision process.

We train the logistic regression Pupil using standard cross-entropy loss and achieve a test accuracy of $91\%$. While this performance is respectable, it falls significantly short of state-of-the-art models on MNIST, and well above the Bayes optimal error rate, which is believed to be below $1\%$ (Mitchell, 1997). In order to improve the performance, we introduce a Tutor model trained to learn minimal, input-level corrections to increase the accuracy of the Pupil's predictions.

To implement the Tutor-Pupil scheme in this setting, we aim to train a Tutor that applies minimal modifications to input images so that the Pupil's performance improves on these corrected versions. However, directly learning pixel-wise corrections in the high-dimensional image space is both computationally expensive and prone to overfitting. To address this, we adopt a strategy inspired by recent advances in generative modeling, which allow for compact and structured representations of images (Bond-Taylor et al., 2021). In particular, we utilize a pretrained variational autoencoder (VAE) (Kingma & Welling, 2014), a widely used generative model that maps high-dimensional data to a lower-dimensional latent space and reconstructs the original inputs from these representations. The VAE regularizes the latent space by enforcing a Gaussian prior, ensuring smoothness and continuity in the learned encoding. This structure enables us to train the Tutor to operate in the latent space, where modifications are more computationally tractable. Figure 5(b) shows the architecture we used for this modification. This structure is functionally equivalent to the general Tutor philosophy shown in Figure 2(b), where the modified input $x' = x + \epsilon$ corresponds to the image reconstructed from the perturbed latent code.

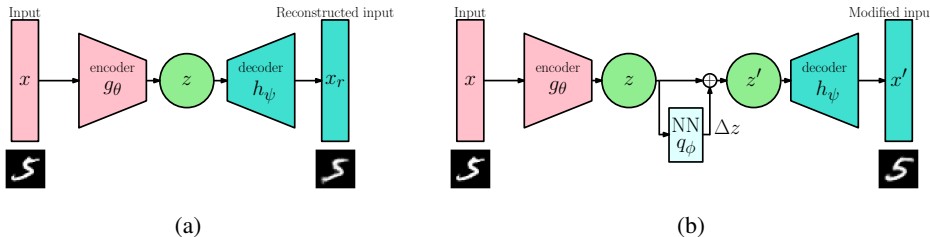

(a)                  (b)

Figure 5: (a) shows the architecture of a variational autoencoder that is trained to encode an image into the latent space and then decode the latent space into a reconstructed image. (b) is the structure of the Tutor model chosen for the image classification example. The neural network $q_\phi$ is designed to modify the latent variable $z$ minimally.

We can train this neural network to minimize the following loss:

$$\mathcal{L} = \underbrace{\mathcal{L}_{\mathrm{C}}(y, f(x'))}_{\text{classification}} + \lambda_1 \underbrace{D_{\mathrm{KL}}\left(q_\phi(z' \mid z) \,\|\, g_\theta(z \mid x)\right)}_{\text{latent consistency}} + \lambda_2 \underbrace{\left\|x' - x\right\|_2^2}_{\text{reconstruction fidelity}}, \qquad (1)$$

where $x$ is the input image and $y$ its true label. The encoder $g_\theta$ maps $x$ to a latent representation $z$. The transformation network $q_\phi$ produces a perturbation $\Delta z$ on the latent vector $z$ yielding $z'$, which is decoded by the pretrained decoder $h_\psi$ to yield $x' = h_\psi(z')$. The logistic regression $f$ then predicts on the modified image $x'$. Importantly, the parameter $\lambda_1$ controls the degree of change in the latent space, enforcing that $z'$ remains close to $z$. This constraint encourages the Tutor to make minimal, interpretable modifications while still improving the Pupil's performance.

The augmented Tutor-Pupil model achieves a test accuracy of $98.5\%$ (vs $91\%$ when only the Pupil model is used), marking a significant improvement over the baseline logistic regression model. Figure 6 highlights several representative examples where the original model fails, but the Tutor-Pupil augmentation yields the correct classification. With a closer examination of these corrected cases, we can see that in almost all of these, the original handwriting deviates from the standard forms, either due to incomplete strokes, irregular proportions, or stylistic variations. We also notice that in all the examples, Tutor subtly modifies the input in a way that restores the key structural features. For example, the Tutor completes the circular loop in a malformed $0$ or $8$ (as shown in Figure 6(a) and 6(h)), or extends a slight vertical stroke to clarify the poorly formed $4$ (Figure 6(d)). These adjustments make the image more legible to the Pupil and allow it to produce the correct classification.

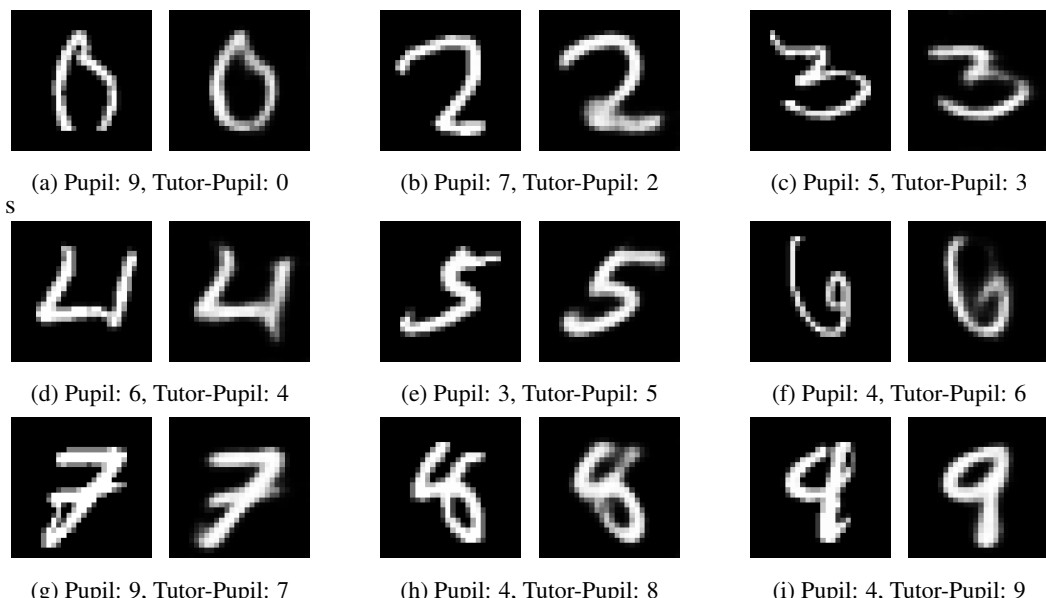

s

(a) Pupil: 9, Tutor-Pupil: 0          (b) Pupil: 7, Tutor-Pupil: 2          (c) Pupil: 5, Tutor-Pupil: 3

(d) Pupil: 6, Tutor-Pupil: 4          (e) Pupil: 3, Tutor-Pupil: 5          (f) Pupil: 4, Tutor-Pupil: 6

(g) Pupil: 9, Tutor-Pupil: 7          (h) Pupil: 4, Tutor-Pupil: 8          (i) Pupil: 4, Tutor-Pupil: 9

Figure 6: Representative examples of misclassified MNIST digits by the Pupil that are successfully corrected by the Tutor. The left image in each pair shows the original input, which the Pupil alone misclassifies. The right image shows the minimally modified version produced by the Tutor, leading to a correct prediction. In many cases, the Tutor enhances or reconstructs specific digit-defining features (such as closing open loops or sharpening strokes), making the digits more recognizable and aligned with canonical representations.

From these observations, we infer that the chosen Pupil, though effective for typical inputs, struggles to generalize when the digit's structure departs from familiar patterns that are most common in the training examples. Interestingly, this insight is consistent with the known limitations of a logisitic regression for this task. As a linear classifier operating directly on pixel intensities, logistic regression does not capture spatial structure or compositional regularities in the way convolutional networks do. It is therefore more vulnerable to distortions, omissions, or stylistic inconsistencies in the data.

This limitation is precisely why we selected logistic regression for this experiment. As a structurally simple and well-understood model, it provides a controlled setting in which we can assess whether the Tutor's corrections reveal meaningful failure modes. The alignment between these corrections and the known weaknesses of logistic regression offers concrete support for the broader applicability of the Tutor-Pupil framework. Even when the Pupil is not inherently interpretable, the behavior of the Tutor can illuminate how the model processes inputs and where it falls short, offering a practical means of interpreting otherwise opaque systems.

Since we offer the Tutor–Pupil architecture also as an interpretability tool, we decided to compare it with SHAP (Lundberg & Lee, 2017), given its prevalence, strong theoretical grounding, and widespread adoption in academic research and real-world deployments. SHAP provides a principled way of attributing importance to input features, and for image data this is often visualized as heatmaps where each pixel is weighted according to its relevance for the model's output. In Appendix D, we report the SHAP outputs for the logistic regression model on the representative images of Figure 6, both for the correct label and the misclassified label. While SHAP offers a quantitative feature attribution, in all these cases the resulting heatmaps are challenging for a human to interpret, especially given the large number of pixels involved. By contrast, the Tutor–Pupil explanation produces corrections that directly modify the input in a human-readable way, providing a more intuitive account of the model's behavior.

## 5  OTHER APPLICATIONS AND CONCLUSION

The Tutor-Pupil architecture proposed in this work offers a novel framework that bridges the gap between performance and interpretability in machine learning systems. When the Pupil is designed

to generalize well (whether it is a model derived from first principles or a learned function), the Tutor acts as a secondary layer that improves upon the Pupil's residual errors and exposes aspects of the underlying phenomenon the Pupil is describing. When the Pupil is a complex or opaque model, the Tutor instead serves as an explanatory layer for the Pupil itself, revealing patterns in how it responds to input data and highlighting the features it relies on to succeed. In both cases, the Tutor is not merely a correction mechanism, but an interpretive tool. Unlike many explanation methods such as counterfactual methods (Prado-Romero et al., 2024), additive explanations (Doumard et al., 2023), example based methods (Van Der Waa et al., 2021), that are purely local (e.g. they focus on an individual input), the Tutor is trained across the entire dataset. As such, it captures global patterns in the Pupil's behavior and failure modes, providing a broader understanding of the system.

While the present work trains the Tutor after the Pupil, a promising future direction is to develop joint training procedures where the two systems evolve simultaneously. For instance, enforcing a form of functional orthogonality between Tutor and Pupil (so that each captures distinct, complementary aspects of the task) may yield both better performance and more disentangled explanations. Moreover, the flexibility of the architecture enables other novel uses. The Tutor can be trained adversarially, not to help but to hinder the Pupil, revealing weak spots or brittle dependencies. This adversarial mode could serve as a diagnostic tool, highlighting vulnerabilities or testing robustness. Another valuable application is bias detection: if the Pupil is influenced by confounding factors in the training data, the Tutor may systematically learn to undo these influences, thereby surfacing implicit biases or spurious correlations.

On the other hand, although interpretability is promoted by encouraging minimal perturbations, the Tutor–Pupil framework remains flexible enough to incorporate additional strategies that further enhance the clarity of learned connections. Depending on the application, this may include constraining the Tutor to operate on superpixels or other human-aligned feature decomposition methods (Ribeiro et al., 2016), thereby improving the interpretability of the resulting explanations.

ACKNOWLEDGMENTS

This work is partially funded by the European Union (Horizon Europe, ERC, COMPLETE, 101075836). Views and opinions expressed are however those of the authors only, and do not necessarily reflect those of the European Union or the European Research Council Executive Agency or the European Commission. Neither the European Union nor the granting authority can be held responsible for them.

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

# A    IMPLEMENTATION DETAILS AND PERFORMANCE METRICS FOR SECTION 2

In order to implement the Tutor-Pupil augmentation as discussed in section 2, we choose a shallow decision tree as our pupil model and fit it to the data. As mentioned previously, in order train a Tutor neural network that will produce the minimal modifications $\epsilon$, we use the following loss function:

$$\mathcal{L} = \underbrace{\mathcal{L}_C(y, f(x + \epsilon))}_{\text{classification}} + \lambda \underbrace{\|\epsilon\|_2^2}_{\text{correction magnitude}}$$

To optimize the weights of the Tutor, we must backpropagate the gradient of the overall loss. Since this loss includes the classification loss from the Pupil, the gradient must flow through the Pupil's function. However, as the Pupil's weights are frozen, this gradient is used solely to compute the derivative of the loss with respect to the parameters of the Tutor. To enable this implementation, we approximate the decision tree with a differentiable function, defined as follows:

$$\theta_1 = \sigma(k \times (x_1 - lower_{x_1})),$$
$$\theta_2 = \sigma(k \times (-x_1 + upper_{x_1})),$$
$$\theta_3 = \sigma(k \times (x_2 - lower_{x_2})),$$
$$\theta_4 = \sigma(k \times (-x_2 + upper_{x_2})),$$
$$f(x) = \theta_1\theta_2\theta_3\theta_4,$$

where $\sigma(.)$ is the sigmoid function function. The parameters $lower_{x_1}$, $upper_{x_1}$, $lower_{x_2}$, and $upper_{x_2}$ are determined by fitting the model to the data. This function works similarly to a decision tree since it checks if the features $x_1$ and $x_2$ lie within the range $[lower_{x_1}, upper_{x_1}]$ and $[lower_{x_2}, upper_{x_2}]$ respectively. The parameter $k$ determines how sharp the edges and corners of these boundaries are enforced by sharpening the sigmoid plot. If the data point is inside the polygon, $\theta_1, \theta_2, \theta_3, \theta_4$ are all equal to 1, if any of them is 0, label 0 is assigned instead ($f(x) = 0$). The polygon fitted to this data is shown in Figure 7(a) with a purple stroke. The parameters found by fitting the model to the data are:$lower_{x_1} = 1.2, upper_{x_1} = 2.87, lower_{x_2} = 1.43, upper_{x_2} = 2.36, k = 110.0$.

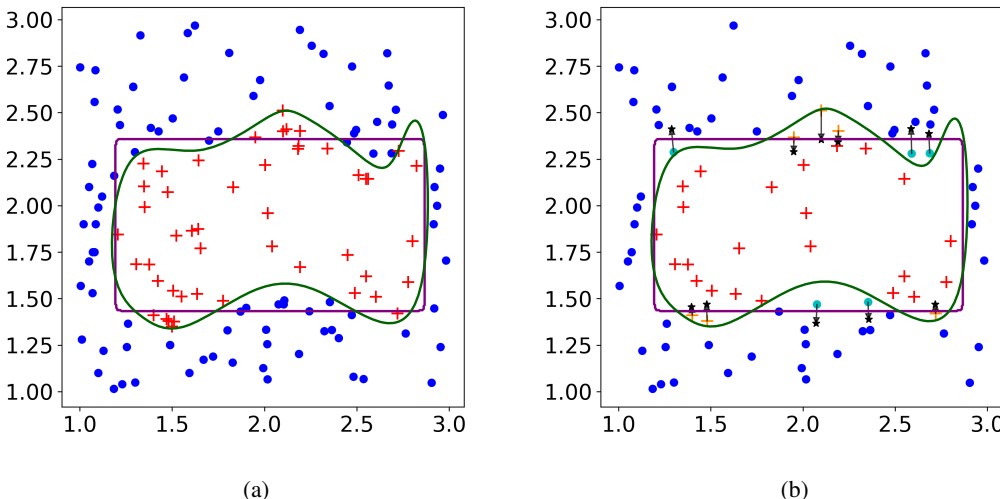

(a)                                            (b)

Figure 7: (a) represents the ground truth boundaries of a binary classification problem. The polygon represents the interpretable function $f$ that performs classification. Data represented in orange and light blue in (b) are data that the augmented model classified them differently than the interpretable model. The minimal change $\epsilon$ identified by the neural network is depicted with a black arrow and the modified data point $x + \epsilon$ is depicted with a black star.

Now $f(x)$ is a differentiable function and we can implement the Tutor. Both the dataset and the complete code are included in the attached ZIP file.

For this toy example, we conduct variability analysis to better understand if the augmentation of the decision tree with the neural network truely leads to better performance metrics.

To evaluate variability, we conducted 6 independent runs of 5-fold cross-validation (a total of 30 training procedures) for the simple model. The best-performing model (based on validation loss) from each run was selected as the Pupil in the Tutor-Pupil framework. We then applied the same procedure for training the augmented model. The accuracy, precision, and Loss Per Sample (LPS) are reported below (mean and std).

Table 1: Accuracy and precision (mean and standard deviation) across conditions.

| Condition | Accuracy (mean) | Accuracy (std) | Precision (mean) | Precision (std) |
|---|---|---|---|---|
| Tree training | 0.8624 | 0.0131 | 0.8459 | 0.0139 |
| Tree validation | 0.8583 | 0.0290 | 0.8385 | 0.0306 |
| Tree test | 0.8588 | 0.0254 | 0.8413 | 0.0415 |
| Augmented training | 0.9768 | 0.0464 | 0.9655 | 0.0716 |
| Augmented validation | 0.9754 | 0.0485 | 0.9610 | 0.0795 |
| Augmented test | 0.9705 | 0.0532 | 0.9512 | 0.0859 |

Table 2: LPS values (mean and standard deviation) across conditions.

| Condition | LPS (mean) | LPS (std) |
|---|---|---|
| Tree training | 0.4217 | 0.0296 |
| Tree validation | 0.4586 | 0.1088 |
| Tree test | 0.4446 | 0.0965 |
| Augmented training | 0.8436 | 0.0965 |
| Augmented validation | 0.9739 | 2.5082 |
| Augmented test | 1.0350 | 2.6602 |

We also evaluated whether the observed improvement in test accuracy for the augmented model is statistically significant. To this end, we performed a one-sided Welch's t-test with the following hypotheses:

$$H_0 : \mu_{\text{tree}} \geqslant \mu_{\text{augmented}},$$
$$H_1 : \mu_{\text{tree}} < \mu_{\text{augmented}}.$$

We set the significance level at $p < 0.05$. The resulting one-sided p-value was $2.648 \times 10^{-11}$, leading us to reject the null hypothesis and conclude that the augmented model achieves a statistically significant improvement in test accuracy.

## B  SIMULATION PROCEDURE FOR SECTION 3

This section describes the simulation procedure for spherical particles of finite radius $r > 0$ confined within a two-dimensional square box of area $V$. A brief but detailed description is given in this section, however the code, given in a zip file, contains mode detailed comments.

Given the box dimension $b$ (assuming a square box so that $V = b^2$), the number of particles $n$, and the particle radius $r$, we initialize the positions $x$ and velocities $v$ of all particles at random, while ensuring that no two particles overlap at the start.

At time $t = 0$, the simulation begins by identifying the earliest collision. This is determined by computing the first time at which the distance between the centers of any two particles becomes less than $2r$. Collisions with the box walls are also considered in this step.

Once the earliest collision is detected, we compute the corresponding collision time $\Delta t$ and identify the collision pair, which may be either two particles or a particle and a wall. The simulation time is then advanced to $t \leftarrow t + \Delta t$, and particle positions are updated according to $x \leftarrow x + v\Delta t$.

The velocities of the colliding pair are then updated. For particle-particle collisions, the update reflects conservation of momentum and energy; for particle-wall collisions, the relevant component of the velocity vector is reflected.

This process is repeated iteratively until the total simulation time exceeds a prescribed maximum duration $T_{\max}$.

### B.1 MEASURING PRESSURE AND TEMPERATURE

To connect macroscopic thermodynamic quantities to microscopic properties of the simulated system, we begin with a fundamental result of the kinetic theory of gases, according to which, the average kinetic energy of a particle is given by:

$$E = \frac{n_d}{2} k_b T. \tag{2}$$

Here $n_d$ denotes the number of degrees of freedom (equal to 2 for a monoatomic particle confined to two dimensions), and $k_b$ is Boltzmann's constant. Independently, the kinetic energy may also be expressed in terms of the mass and average velocity of a particle:

$$E = \frac{m}{2} v_{\text{average}}^2, \tag{3}$$

where $m$ is the particle mass, and $v_{average}$ is the average velocity of particles. By equating these two expressions and solving for $T$, we obtain:

$$T = \frac{m}{n_d k_b} v_{average}^2. \tag{4}$$

Thus, just by appropriately rescaling the temperature by a scalar constant, with no loss of generality, we can define it directly in terms of the average kinetic energy of the particles, which is computed as the mean of the squared norms of their velocity vectors.

The pressure may be computed by invoking the kinetic theory of gases in its two-dimensional formulation, which relates macroscopic force to microscopic momentum exchanges.

$$P = \frac{F}{4b}. \tag{5}$$

Here, $F$ denotes the total force exerted by the particles on the walls of a square container of width $b$. This force arises from the cumulative effect of momentum changes due to collisions with the boundaries. According to Newtonian mechanics, force is defined as the rate of change of momentum:

$$F = \frac{\Delta p}{\Delta t}, \tag{6}$$

where $\Delta p$ is the total change in momentum due to particle-wall collisions, defined as

$$\Delta p = \sum_i |\Delta p_i|, \tag{7}$$

with each $\Delta p_i$ representing the momentum change from an individual collision. The sum is taken over all collisions during the simulation, and the absolute value ensures that all impulses, regardless of direction, contribute positively to the total momentum transfer. Since momentum is given by $mv$, and changes in momentum arise from velocity reversals upon wall impact, we compute each $\Delta p_i$ as $m|\Delta v_i|$, where $\Delta v_i$ is the change in the velocity component perpendicular to the wall. Summing these over all collisions yields the total momentum transfer. The resulting average force is then given by

$$F = \frac{\Delta p}{T_{\text{max}}}, \tag{8}$$

where $T_{\text{max}}$ is the total simulation time. The measured pressure is thus given by:

$$P = \frac{\frac{\Delta p}{T_{\text{max}}}}{4b} \tag{9}$$

### B.2 TUTOR IMPLEMENTATION

The simulations were performed for 15 different volume settings and 8 different temperature settings, resulting in a total of 120 data points.

The Pupil model is based on the ideal gas law, which estimates pressure using the equation $\hat{P}_i = \frac{nRT}{V}$. This model is implemented in PyTorch.

Next, we construct the Tutor model, also in PyTorch. It takes temperature $T$ and volume $V$ as input and outputs a correction vector $\epsilon = (\epsilon_V, \epsilon_T)$. The model consists of two hidden layers and uses

a final activation function of `Tanh()` to ensure that the corrections remain bounded and small in magnitude.

However, because the simulated values of $V$ and $T$ lie in the ranges $V \in [13550.0, 88492.0]$ and $T \in [60.0, 1224.0]$, respectively, we scale the output of the `Tanh()` function accordingly. We then create an augmented model as illustrated in Figure 2(b).

## C    IMPLEMENTATION DETAILS FOR SECTION 4

In this section, we briefly go over the implementation of the Tutor-Pupil framework discussed in Section 4. The Pupil is a logistic regression model, implemented in PyTorch as a single linear layer. It is trained using the cross-entropy loss and optimized with Adam for 10 epochs, achieving a test accuracy of $91\%$.

To construct the Tutor, we employ a variational autoencoder (VAE) that operates in a compact latent space rather than directly in the input space. While pretrained VAEs are available through libraries such as PyTorch and TensorFlow, they are typically designed for high-resolution or complex images, which makes them suboptimal for our case. Our data consists solely of handwritten digits with a resolution of $28 \times 28$ pixels, so we opt to train a smaller VAE tailored to this setting.

The encoder of our VAE comprises three convolutional layers followed by a linear layer that outputs the mean $\mu$ and the log-variance $\log(\sigma^2)$ of the latent distribution. The latent space is set to have dimension 32. The decoder consists of a linear layer followed by three transposed convolutional layers. During training, an input image is passed through the encoder, a latent variable $z$ is sampled from the distribution $\mathcal{N}(\mu, \sigma^2)$ using the reparameterization trick, and the decoder reconstructs the image from $z$.

The VAE is trained for 50 epochs using the Adam optimizer. The loss function used is the standard variational autoencoder loss, which combines a reconstruction term with a regularization term enforcing the approximate posterior to match a standard normal prior. Formally, the loss function is given by

$$\mathcal{L}_{\text{VAE}}(x) = \mathbb{E}_{q_\phi(z|x)}\left[\log p_\theta(x|z)\right] - D_{\text{KL}}\left(q_\phi(z|x) \,\|\, p(z)\right), \tag{10}$$

where $q_\phi(z|x)$ is the encoder (approximate posterior), $p_\theta(x|z)$ is the decoder (likelihood), and $p(z) = \mathcal{N}(0, I)$ is the prior over the latent variables.

Since VAEs are known to smooth image features, we evaluated the effect of applying VAE encoding and decoding to the images without involving the Tutor. Under this setting, the logistic regression model achieved an improved accuracy of $91.7\%$.

The Tutor is then implemented to apply minimal change to the latent variable $z$ such to minimze the loss given in equation equation 1.

This network consists of four sequential linear layers. The complete augmented model is constructed by combining the Pupil (logistic regression model), the Tutor network, and the pretrained VAE.

After initialization, the weights of both the VAE and the Pupil are frozen. When an image is input to the augmented model, the encoder of the VAE produces the mean $\mu$ and the log-variance $\log(\sigma^2)$. The mean $\mu$ is then fed to the Tutor network to produce $z'$ in order to eventaully produce $x'$ via the decoder.

The reconstructed input $x'$ is subsequently passed to the Pupil model, which outputs a predicted label $y$.

The Tutor network is trained for 40 epochs using the Adam optimizer to minimize the loss defined in Equation equation 1. The hyperparameters $\lambda_1$ and $\lambda_2$ are selected based on empirical tuning. The final test accuracy of the augmented model reaches $98.57\%$. This outcome demonstrates a substantial improvement over the baseline performance obtained by applying the Pupil model directly to the original input $x$.

We note that this performance depends critically on the quality of the latent space representation. In particular, replacing the VAE with a simple autoencoder would significantly reduce the likelihood that the full augmented model will perform well. This is because VAEs employ the reparameterization trick, which enables sampling of the latent variable via $z = \mu + \sigma \cdot \delta$, where $\delta \sim \mathcal{N}(0, I)$, in a

way that permits gradient propagation during backpropagation. This mechanism ensures that small changes in the input $x$ produce small and consistent changes in the latent representation $z$.

As a result, the latent space learned by the VAE is smooth, continuous, and structured. Furthermore, because the latent variables are regularized toward a standard Gaussian distribution, interpolations between latent vectors tend to remain within regions the decoder has been trained to reconstruct accurately. This property enables the generation of semantically meaningful outputs across the latent space. In contrast, traditional autoencoders, which lack such regularization, often produce unstructured latent spaces where interpolated points may correspond to implausible or out-of-distribution inputs.

## D   COMPARISON WITH SHAPLEY VALUES FOR SECTION 4

In this section we provide a comparison between the explanation given by the application of the Tutor-Pupil architecture for the logistic regression model applied to the MNIST dataset and the explanations obtained with SHAP, a widely used method in explainable AI,

The results are reported in Figure 8 for some representative MNIST examples corresponding to cases of misclassification by the Pupil acting alone. Each subfigure is organized as a 3×2 grid, with the original image (top left), its VAE-based reconstruction (top right) obtained with the help of the Tutor, and two SHAP heatmaps corresponding to the correct label (middle row) and to the label predicted by the Pupil alone (bottom row). The intent is to highlight the difference between traditional feature-attribution explanations and the more direct, human-readable corrections provided by the Tutor. While SHAP offers pixel-level relevance scores, the resulting heatmaps are often difficult to interpret: they indicate where the Pupil focuses in the image, but they do not provide a clear sense of how the model is reasoning or why it fails. For instance, in misclassified cases, the Pupil's heatmap of attribution scores remains diffuse and not easily translatable into an intuitive explanation of the classification error. By contrast, the Tutor–Pupil approach explicitly modifies the input in a minimal and targeted way, effectively "adding" the features that the Pupil requires to reach the correct label. These corrections are directly visual and semantically meaningful (for example, sharpening the loop of a "0" or extending the top bar of a "7") and thereby offer a much clearer view into the Pupil's internal logic and failure modes. In this sense, the Tutor does not merely localize important pixels but provides a global, structured intervention that is accessible to human interpretation, complementing and often surpassing the insights obtainable from standard attribution methods like SHAP.

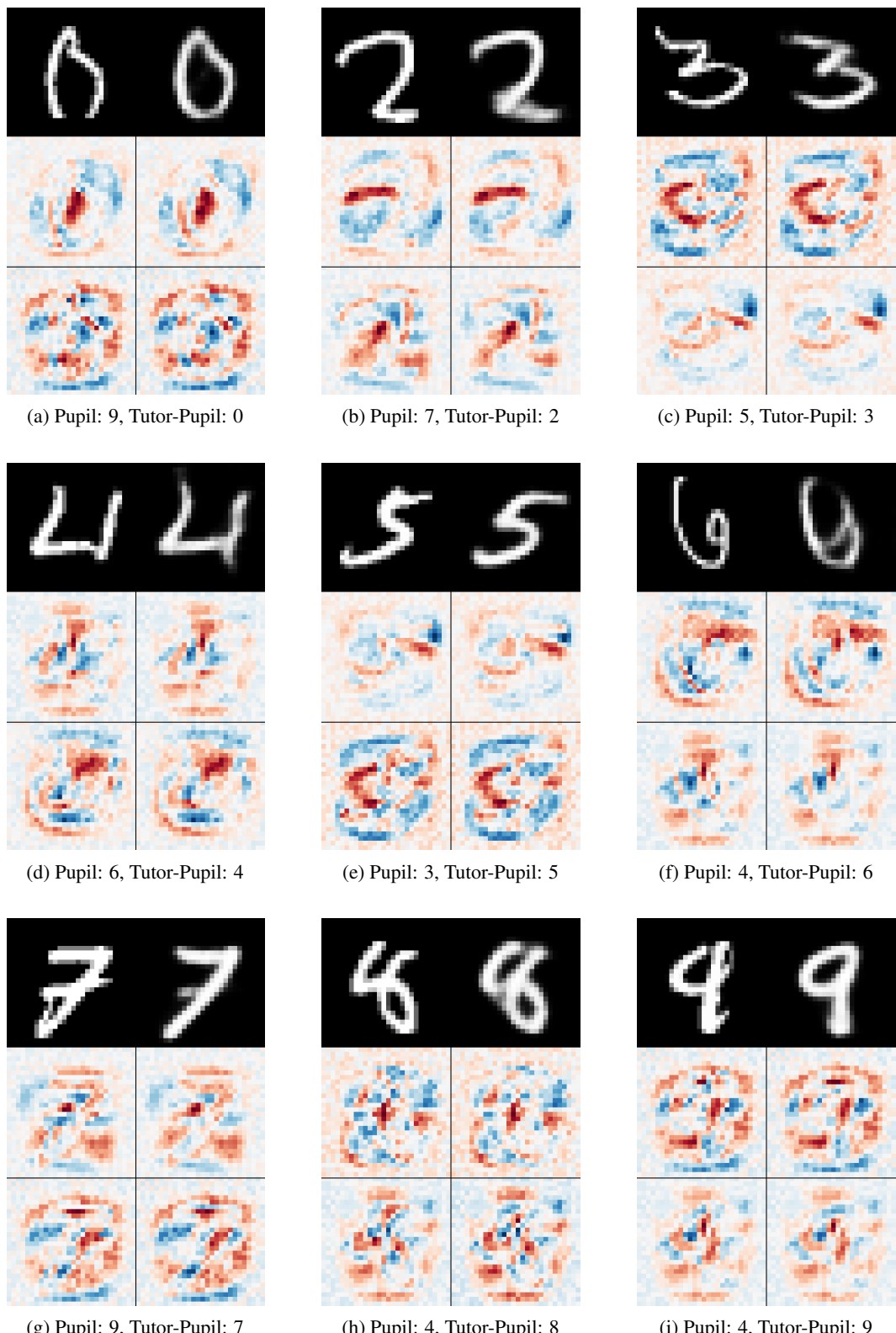

(a) Pupil: 9, Tutor-Pupil: 0

(b) Pupil: 7, Tutor-Pupil: 2

(c) Pupil: 5, Tutor-Pupil: 3

(d) Pupil: 6, Tutor-Pupil: 4

(e) Pupil: 3, Tutor-Pupil: 5

(f) Pupil: 4, Tutor-Pupil: 6

(g) Pupil: 9, Tutor-Pupil: 7

(h) Pupil: 4, Tutor-Pupil: 8

(i) Pupil: 4, Tutor-Pupil: 9

Figure 8: Each subfigure shows a 3×2 grid with original image (top left), decoded image (top right), SHAP explanations for correct label (middle row), and SHAP explanations for label output by the Pupil (bottom row).

