# OpenReview forum: "The Tutor-Pupil Augmentation: Enhancing Learning and Interpretability via Input Corrections"
_ICLR.cc/2026/Conference — ICLR 2026 Poster_

### Official Review · Reviewer_4yEq · 2025-10-15

**Soundness:** 2
**Presentation:** 2
**Contribution:** 2
**Rating:** 4
**Confidence:** 2

**Summary:**

This paper addresses the long-standing "performance-interpretability trade-off" in machine learning—where simple, interpretable models (e.g., decision trees, physics-based formulas) lack expressive power, while complex black-box models (e.g., neural networks) sacrifice transparency. It proposes a novel Tutor-Pupil augmentation framework to resolve this trade-off by leveraging "minimal input-space corrections" rather than output adjustments, enabling both performance gains and enhanced interpretability.

**Strengths:**

- Unlike prior work that corrects outputs (e.g., residual networks, ensemble stacking), this paper corrects inputs, preserving the Pupil’s interpretability.

- The paper’s originality lies in redefining the paradigm of model augmentation, removing limitations of prior work, and creating novel links between data-driven learning and theoretical insight—all of which challenge long-standing practices in interpretable AI.

- The paper does not limit the Tutor-Pupil framework to a single task type but adapts it to three distinct domains—a creative extension that proves its generality.

**Weaknesses:**

- The paper strictly adopts a "train Pupil first, then train Tutor" serial paradigm (Pupil parameters are frozen during Tutor training but fails to explore joint training—a critical gap that limits the framework’s ability to fully leverage synergies between the two models and may amplify Pupil’s inherent flaws.

- Novelty Gap: “Input-space correction” is not new. Position the paper as “systematic, global counterfactuals for interpretable models” rather than a brand-new paradigm and provide a taxonomy table that shows how Tutor-Pupil differs from (i) local counterfactuals, (ii) adversarial examples, (iii) data-augmentation policies on objectives, constraints, and evaluation metrics.

- The paper validates the framework exclusively with interpretable Pupils (decision trees, logistic regression, ideal gas law but fails to test black-box Pupils (e.g., ResNet, Transformer)—a critical gap, as many real-world systems rely on complex models that need interpretive tools (e.g., medical image classifiers using CNNs).

**Questions:**

see weakness

---

> ### Author Response · Authors · 2025-11-25
> **authors' first response**
>
> We thank the reviewer for their time and review and their positive comments.
> We address the weaknesses here.
>
> ### "The paper strictly adopts a "train Pupil first, then train Tutor" serial paradigm (Pupil parameters are frozen during Tutor training but fails to explore joint training—a critical gap that limits the framework’s ability to fully leverage synergies between the two models and may amplify Pupil’s inherent flaws."
>
> Yes, it is indeed possible to train the neural network and the interpretable model together, and this represents an interesting research direction with significant potential.
> In parallel model augmentation, literature has explored both approaches: fixing the interpretable model or training it jointly with the neural network, and each comes with its own advantages and challenges [Györök 2025].
> For example, it has been shown that in simultaneous training, the neural network can sometimes overpower the interpretable model if not carefully controlled, although strategies to mitigate this are actively being investigated.
>
> Additionally, when the interpretable model is based on first principles, there is limited consensus on how to interpret the results of joint training with a neural network. In our work, we chose to separate the training: we first optimize the fixed model to capture the data as accurately as possible (or take it from first principles) and then train the tutor to improve its performance.
> While leveraging potential synergies through joint training is appealing, we felt it could initially complicate interpretation and the message.
>
> Nevertheless, we acknowledge that joint training is a promising direction, particularly given recent advances in the field.
> On the other hand, we deem it unlikely that with separate training the Tutor might "amplify the Pupil's inherent flaws" as the Reviewer hypothesizes: during the optimization the Tutor has always the option to leave the the input unmodified and a regularization parameter keeps the modifications small precisely to avoid this.
>
>
> ### ''Novelty Gap: “Input-space correction” is not new. Position the paper as “systematic, global counterfactuals for interpretable models” rather than a brand-new paradigm and provide a taxonomy table that shows how Tutor-Pupil differs from (i) local counterfactuals, (ii) adversarial examples, (iii) data-augmentation policies on objectives, constraints, and evaluation metrics.''
>
> We thank the reviewer We thank the reviewer for this question.
> Our framework is a model augmentation approach: the Tutor learns an input-space modification
> $g(x)$ that becomes part of the predictive pipeline and is optimized to improve the Pupil's performance.
>
>
> Counterfactual methods, in contrast, are designed to explain a decision and do not aim to improve predictive accuracy.
> For a specific instance $x$, these methods simply search for the closest $x'$ that changes the model's decision. Such a search is performed independently for every data point of interest.
>
> The Tutor instead learns a global mapping
> $g$ over the entire dataset by optimizing a loss that trades off performance improvement and the size of the modification. Since
> $g(x)$ is learned jointly rather than independently for each instance, it provides a more holistic view of which input adjustments are broadly useful and potentially capable to learn more general modification rules.
>
> Adversarial examples similarly generate instance-specific perturbations, but with the explicit objective of inducing incorrect predictions to reveal model vulnerabilities. Thus, as in standard counterfactual methods the explanation is exclusively local.
>
>
> Data-augmentation strategies create transformed versions of the input to increase training diversity. They modify the dataset before training and do not learn corrections tied to model performance at inference.
>
> Following the reviewer’s suggestion, we will include a taxonomy table that compares Tutor–Pupil with local counterfactuals, adversarial examples, and data-augmentation policies along the axes of objective, type of modification, constraints, and evaluation criteria.
>
> We thank the reviewer again for these constructive comments.

---

> > ### Author Response · Authors · 2025-11-25
> > **author's first response**
> >
> > ### ''The paper validates the framework exclusively with interpretable Pupils (decision trees, logistic regression, ideal gas law but fails to test black-box Pupils (e.g., ResNet, Transformer)—a critical gap, as many real-world systems rely on complex models that need interpretive tools (e.g., medical image classifiers using CNNs).''
> >
> > We thank the reviewer for this comment.
> > We see this point differently, and we take the opportunity to clarify our rationale for the chosen experiments.
> >
> >
> > The illustration via decision trees is to show that we can keep an interpretable model (the Pupil) and still boost its performance by augmenting it with a Tutor.
> > Interpretability is kept by keeping the corrections of the Tutor limited.
> > In this illustration, it wouldn't be useful to have an uninterpretable Pupil: if both Pupil and Tutor are uninterpretable we could have used a single uninterpretable model.
> >
> > The ideal gas law is an illustration of how the Tutor can be used to obtain more accurate analytical models from first principle models (the law of ideal gases) and potentially even derive "higher order principle models" (e.g. van der Waals law for real gases). The tutor is not an interpretable tool in this case. Instead, symbolic regression or other tools could be applied to the Tutor to obtain more sophisticated explicit analytical expressions. Even in this case having a Pupil that is not interpretable would not help to reach this goal since we wouldn't be able to improve on the first principle models.
> >
> > The last example is about non-interpretable pupils. The reviewer states that a logistic regression is interpretable. We have a different perspective. The obtained logistic regression has a number of parameters that is large relative to the number of parameters in a model that a human interpreter is capable of processing to evaluate if the model is performing its function properly.
> > That is why we state that Tutor can be seen as a tool for interpretability as well, since in this case it is capable of highlighting which components of the image are influencing the decision in a way that a human interpreter can understand. Other prevalent interpretability methods, such as SHAP, are definitely not able to provide these insights with equal ease as exemplified in the appendix.
> >
> > Large-scale examples are certainly an important direction we intend to explore with this framework. However, the examples presented in the paper were deliberately kept simple so that the operation of the Tutor–Pupil mechanism would remain fully transparent, without the confounding effects that more complex datasets might introduce:
> > the goal was to showcase the advantages of the framework with examples that can be fully inspected.

---

> > > ### Comment · Reviewer_4yEq · 2025-11-27
> > >
> > > Thank you for the response. The authors have addressed most of my questions. I have raised my score from 4 to 6. I will further discuss with the other reviewers.

---

### Official Review · Reviewer_YX96 · 2025-10-29

**Soundness:** 2
**Presentation:** 3
**Contribution:** 2
**Rating:** 2
**Confidence:** 4

**Summary:**

In this paper, the auxiliary model in the model augmentation framework is used to boost prediction accuracy and interpretability. In a Tutor-Pupil scheme, the Pupil is used to learn according to the domain-specific features, while the Tutor adding a small perturbation to the input of Pupil, is used to uncover the specific failure modes and regions of uncertainty with the Pupil's predictions. In physics-informed models, higher-order global and structural information in the data space and decision boundaries is revealed leading to better modeling of the observations and the explanation of the shortcomings of the reference dynamical processes.

**Strengths:**

The paper is well written and easy to follow. The authors provide a sound a good background for the problem of interpretability-complexity trade-off of the modern architectures used in ML.

The authors propose a Pupil-Tutor framework to boost the performance and interpretability of simple architecture or numerical models. This method works for physical models and can be used to modify them according to the observations for better understanding of the underlying processes.

The authors provide other examples in the image datasets where the Tutor improves the prediction performance of a non-interpretable using this architecture.

**Weaknesses:**

The scope of interpretability of the Tutor is limited to our understanding of the perturbations in the data space. The results are constrained to synthetic 2d data, 3 variable time independent physical system, which are too simple to understand the potential of the proposed architecture, and logistic regression on MNIST dataset which in I believe had vague inconclusive results on diagnostics and interpretability of the failure modes of a non-interpretable model.

This might be due to the nature of the Pupil perturbations that are performed in data space and therefore the interpretability is left to human understanding of the discernible features in the data space, which is another complex task. While portrayed as a tool for interpretability, I believe this framework is yet ineffective and cannot improve interpretability as well as e.g. [Sarvmaili'24], where a set of representative samples are produced which can be used to understand the main modes of failure from training data. The authors don't report any evaluation metrics on the interpretability of the predictions especially for high dimensional datasets. A more comprehensive study can be performed e.g. with 3dshapes dataset where the main features are clearly discernible in the data space, or applying known transformations to the MNIST dataset and training the Tutor to undo said transformations could lead to more conclusive results.

[Sarvmaili'25] Data-centric Prediction Explanation via Kernelized Stein Discrepancy, Sarvmaili, Mahtab and Sajjad, Hassan and Wu, Ga, ICLR 2025

**Questions:**

Have the authors checked the robustness of the Pupil-Tutor framework?

Famously, [Goodfellow'14] showed that the addition of a small but unstructured noise to the input leads for the change in the class, but not a visible difference in the image itself. In comparison addition of the noise in the latent space would likely translate to a visually significant changes in the image after decoding, but may still not be interpretable as the encoder compresses the image through entangling the meaningful features. As the results of the MNIST experiment are not very conclusive to me, can the authors explain how the Tutor perturbations are interpretable and different from a random cohesive structured blob?

---

> ### Author Response · Authors · 2025-11-25
> **authors' first response**
>
> We thank the reviewer for the comments and the provided references.
> We understand the main concerns are twofold: (i) the fact that interpretability/explainability is left to humans and the need for quantitative metrics to evaluate interpretability/explainability and its robustness and (ii) the limited scope of high-dimensional experiments, with the MNIST results seen as insufficiently definitive.
>
> First, we would like to emphasize that our framework is not solely an interpretability method.
> It is an augmentation framework designed to improve the performance of a pupil model while also providing the additional benefit of preserving/enhancing interpretability/explainability
> Because of this dual role, the framework applies to a broad range of scenarios as we have illustrated through several distinct examples.
>
>
> About concern (i), Interpretation and explanation are inherently complex concepts, and as a result, there is no single universally accepted definition for them.
> Moreover, what counts as a meaningful interpretation/explanation could often depend on the application or domain (Buijsman et al. 2022; Beisbart \& Raz 2022; Paez et al. 2019; Gilpin et al. 2022). This has naturally led to the development of different families of methods. For example:
>
> - Feature attribution methods (SHAP, Deeplift, etc) quantify each feature's contribution to a prediction.
> - Counterfactual methods describe how slight input changes could alter the outcome.
> - Example-based methods (e.g., prototype or influential example methods) illustrate model behavior through relevant data points.
>
>
> Due to the fundamental differences between these categories of methods, heterogeneous forms of outputs are expected, and evaluation metrics appropriate for one method may not be meaningful for another.
> At the same time, combining multiple explanation/interpretation approaches can provide human experts with complementary perspectives, helping to verify model reliability and better understand complex interactions.
>
>
> The method suggested by the reviewer (Sarvmaili 2024) is an example-based  approach that uses kernel similarity to identify influential training samples.
> We note, however, that interpretation of its outputs still relies on human judgment.
> The article points out that evaluation of interpretability methods is often qualitative and introduces Hit Rate and Coverage to provide quantitative assessment.
> These metrics, however, are specific to example-based methods.
>
> In contrast, our model augmentation approach is primarily concerned with improving performance, whether the task involves classification or prediction, with the added benefit of enhanced interpretability compared with parallel augmentation.
> This interpretability is more closely aligned with counterfactual explanations (despite some fundamental differences with counterfactual explanations that we highlighted) than with example-based methods.
> While universally accepted metrics for counterfactual explanations do not exist, some studies report metrics such as proximity, validity, and sparsity [Guidotti 2020].
>
> We plan to include these metrics in the revised manuscript, as our method is specifically designed to minimize proximity and maximize validity through performance improvement of the pupil model.

---

> > ### Author Response · Authors · 2025-11-25
> > **authors' first response**
> >
> > With the discussion of different explanation types in mind, we would like to clarify the MNIST example. It is true that interpretability often relies on human understanding of discernible features, as is common in many interpretability tasks [Narayanan et al,.2018].
> >
> > To clarify this example, we can consider a scenario where an example is misclassified by the pupil model but correctly classified by the augmented model, yet there are no visible differences between the original and altered images.
> > This would suggest that the pupil model had learned superficial or irrelevant features that our augmentation helps to bring to light.
> >
> > On the other hand, the results provided in the article indicate that the modification applied to inputs were somewhat aligned with human intuition.
> > From our perspective, this makes sense: logistic regression is a relatively simple model for pixel-level image classification and should capture the most common patterns in the training data, such as fully formed strokes in digits.
> > The analysis of the occurrences where the tutor "corrects" the pupil supports this conclusion.
> >
> > This is not to suggest that our framework will fully replace other explanation methods. Rather, we believe it has potential in applications where experts may prefer this approach, as we have aimed to illustrate with the examples provided in the article.
> >
> > We acknowledge that the examples are simple, but this simplicity is intentional. Our goal was to showcase and verify the usefulness of the framework. Using more complex examples would have required more specialized expertise that not all potetial readers/users might have, and also it would have made it more difficult to reliably assess the interpretations of our examples.
> >
> >
> >
> > Regarding the comment “Famously, [Goodfellow 2014] showed that the addition of a small but unstructured noise to the input leads to a change in the predicted class, without a visible difference in the image itself. In comparison, addition of noise in the latent space would likely produce visually significant changes after decoding, but may still not be interpretable as the encoder compresses the image through entangling meaningful features.”
> >
> > It is indeed important to check that the VAE is not adding any additional performance by smoothing out the features on its own.
> > We found that the architecture [image --> encoder --> latent z --> decoder --> classifier --> y], without the Tutor's corrective effect, increases the accuracy minimally from 91 --> 91.5.
> > We will include this clarification in the article. We hope we have correctly understood the point raised by the reviewer; if not, we would appreciate further guidance.
> >
> >
> > ### ``Have the authors checked the robustness of the Pupil-Tutor framework?''
> >
> > We have. We designed a statistical test framework to see if under different random seeds the performance metrics of Tutor-Pupil augmentation exceeds that of Pupil. We found significant results and we reported them for the first example. We will add it for the rest of examples as well in the revised version.
> > Although we must admit that robustness does not always have an agreed up on definition. If the reviewer has a different definition in mind, we'd be happy to set up an experiments to measure it in additional ways suggested by the reviewer.
> >
> > In the end, we'd like to thank the reviewer for their time and helpful comments. We hope we have clarified all the points and concerns raised by the reviewer.

---

### Official Review · Reviewer_Z1yU · 2025-11-01

**Soundness:** 3
**Presentation:** 4
**Contribution:** 3
**Rating:** 8
**Confidence:** 4

**Summary:**

This paper introduces the Tutor–Pupil augmentation framework, a general approach to improving both model performance and interpretability through minimal input-level corrections. The Pupil is a fixed, interpretable or task-specialized model (e.g., decision tree, physical law, logistic regression), while the Tutor is a flexible neural network trained to apply small corrections to the inputs such that the Pupil produces more accurate outputs. This setup enables the Tutor not only to enhance predictions but also to diagnose failure modes of the Pupil by revealing where and how input perturbations correct errors. The framework is demonstrated on three diverse cases: (1) augmenting a decision tree for a toy binary classification problem, (2) refining the ideal gas law to account for non-ideal behaviors—discovering van der Waals-like corrections, and (3) improving handwritten digit classification via a VAE-based latent-space Tutor acting on a logistic regression Pupil, which also provides interpretable visual corrections. The results show notable gains in accuracy and interpretability across tasks, with meaningful parallels to symbolic regression and explainable AI.

**Strengths:**

The paper is conceptually novel and elegant, offering a unified and interpretable augmentation scheme applicable across interpretable and black-box models. Its theoretical framing—training a Tutor to apply minimal, semantically meaningful corrections—is both intuitive and powerful. The work’s breadth, spanning interpretable (decision trees), physics-based (ideal gas law), and data-driven (MNIST classification) settings, convincingly demonstrates generality. The analyses are rigorous, supported by visualizations. The MNIST experiment is particularly compelling: the Tutor’s subtle adjustments (e.g., closing loops or clarifying strokes) both enhance performance ( and produce human-readable explanations that outperform conventional attribution maps. The idea of deriving physically meaningful corrections from learned input perturbations is especially original and promising for scientific ML applications.

**Weaknesses:**

In the MNIST setting, the performance jump (91%→98.5%) could partly result from the use of a VAE-trained latent representation rather than purely from the Tutor’s corrective effect.

Additionally, the paper could better differentiate its contributions from related ideas like counterfactual explanations, residual learning, and gradient-based input attribution methods.

Finally, the experiments, while creative, are small-scale; a larger empirical evaluation would strengthen the claims of robustness and general utility.

**Questions:**

How sensitive are the Tutor’s corrections to hyperparameters such as λ (correction magnitude regularization)?

Could the framework handle non-differentiable Pupils (e.g., rule-based systems) at scale?

Does the learned correction vector generalize across data distributions or must it be retrained for each Pupil or dataset?

How can one quantify interpretability improvements beyond visual inspection (e.g., through user studies or explanation fidelity metrics)?

Could the Tutor–Pupil setup be extended to adversarial tutoring, where the Tutor exposes brittleness or bias in the Pupil rather than assisting it?

---

> ### Author Response · Authors · 2025-11-25
> **Authors' first response**
>
> We thank the reviewer for the time and effort they put into reading and evaluating our article, and for their detailed and well-thought review.
> We were encouraged by their comments on the variety of our examples, and in particular by their appreciation of our physics-based example and their view of it as a promising direction.
> The reviewer raises fair and insightful points, which we address in detail below.
>
>
> ### `` In the MNIST setting, the performance jump (91→98.5) could partly result from the use
> of a VAE-trained latent representation rather than purely from the Tutor’s corrective effect."
>
>
> This is an important point. Although, we hadn't mentioned in the article, we had checked for this.
> We found that the architecture [image --> encoder --> latent z --> decoder --> classifier --> y] without the Tutor's corrective effect increases the accuracy from 91 --> 91.5.
> We  will report this in article for clarification.
>
>
> ### `` Additionally, the paper could better differentiate its contributions from related ideas like counterfactual explanations, residual learning, and gradient-based input attribution methods.''
>
> We thank the reviewer for raising this point. We agree that differentiating our contributions from related approaches is important and will include a dedicated discussion in the revised paper. Below, we provide a sketch of this contextualization, which will be integrated with other related work suggested by the reviewers.
>
> Our framework is a model augmentation approach: the Tutor learns an input-space modification
> $g(x)$ that becomes part of the predictive pipeline and is optimized to improve the Pupil's performance.
>
> Counterfactual methods, in contrast, are designed to explain a decision and do not aim to improve predictive accuracy.
> For each instance $x$, they search for the closest $x'$ that changes the model's decision, and this search is performed independently for every data point of interest.
>
> The Tutor instead learns a global mapping
> $g$ over the entire dataset by optimizing a loss that trades off performance improvement and the size of the modification. Since
> $g(x)$ is learned jointly rather than independently for each instance, it provides a more holistic view of which input adjustments are broadly useful.
>
> gradient-based attribution methods do not produce modified inputs. Instead, their goal is to quantify the impact of each feature on a single decision. No global mapping  is learned across the dataset, and the objective is fundamentally different from Tutor–Pupil.
>
>
> Different variants of residual learning methods exist.
> Some resemble the parallel augmentation framework already discussed in our article. Others, such as residual connections in ResNet, follow the form:
> $y_{l+1} = y_{l} + F(y_{l})$, where $F(y_l)$ is a learned residual mapping at layer $l$.
>
> These residuals operate in the hidden or feature space to improve gradient flow and ease the training of deep networks. Unlike the Tutor, they do not modify the input itself and are not designed to provide interpretable corrections.
>
>
> ### ''Finally, the experiments, while creative, are small-scale; a larger empirical evaluation would strengthen the claims of robustness and general utility.''
>
> We agree with the reviewer that additional large-scale examples would strengthen the paper, and we are actively working on such extensions. However, the examples included in this work were deliberately simple. Our goal was to showcase the advantages of the framework with examples that can be fully verified. Applying the approach immediately to complex domains, such as medical data or deep neural networks, would have introduced additional complexity that could obscure the validation of the method itself.
>
> ### ''How sensitive are the Tutor’s corrections to hyperparameters such as (correction magnitude regularization)?''
>
> We conducted a preliminary sensitivity analysis to perform sanity checks and to fine-tune the hyperparameters. The results indicate that the Tutor’s corrections vary modestly when the regularization parameter is changed by an order of magnitude, suggesting that the method is not overly sensitive to this hyperparameter. We will include a more detailed report of these results in the appendix of the revised paper.
>
> ### ''Could the framework handle non-differentiable Pupils (e.g., rule-based systems) at scale?''
>
> Not at the moment, this would require different optimization frameworks beyond the classical backprop automatic differentiation setting.
>
> ### ''Does the learned correction vector generalize across data distributions or must it be retrained for each Pupil or dataset?''
>
> This is an interesting question.
> We think if the tutor is trained on a broader spectrum of tasks, some form of transfer learning may indeed be possible across different data distributions or pupils.

---

> > ### Author Response · Authors · 2025-11-25
> > **authors' first response**
> >
> > ### ''How can one quantify interpretability improvements beyond visual inspection (e.g., through user studies or explanation fidelity metrics)?''
> >
> > This is a valid point, and one that the entire interpretability and explainability literature continues to struggle with. Our framework addresses this in two complementary ways:
> >
> >
> > 1. Interpretability is encouraged through the *minimality* of perturbations.
> > A central principle of our method is that the Tutor is trained to modify the input minimally while improving the Pupil's prediction.
> > If minimal scattered perturbations are sufficient to alter the Pupil's decision, for example, barely noticeable pixel changes in an image, this still reveals something meaningful: the Pupil may suffer from robustness issues or may rely on spurious features. Thus, minimality may act both as a regularizer and as an interpretability constraint.
> >
> > 2. Interpretability can be enforced through *structured perturbation spaces depending on the domain.*
> > Since the Tutor-Pupil framework is quite general, different applications allow different ways of constraining the Tutor's actions, as we tried to showcase in some of our examples:
> >
> > - In the ideal gas law example, the symbolic-regression to obtain a more general model step was applied without using any human interpretation of the corrections.
> > We deliberately selected an example where the perturbations were inherently interpretable so that we could verify that the Tutor-Pupil framework behaved sensibly. The resulting data-driven analytic correction matched van der Waals-type terms, providing reassurance that the framework uncovers meaningful structure even without injecting the physical insights that led to the formulation of van der Waals law.
> > - In the MNIST example, we operate in the latent space of a pretrained VAE. This low-dimensional, smooth manifold might naturally preserve semantic structure, helping the decoded perturbations remain visually interpretable.
> > - In other applications, interpretability could instead be enforced through pixel segmentation, superpixels, or other human-aligned feature decompositions [Ribeiro 2016]. The framework is flexible enough to incorporate these constraints.
> >
> >
> > In summary, the Tutor is not free to apply arbitrary corrections: interpretability arises from (i) enforcing minimal perturbations and (ii) constraining the perturbation space in domain-appropriate ways.
> > We agree that this point deserves clearer emphasis. In the revised manuscript, we will highlight this aspect explicitly by including a discussion of how interpretability constraints can be incorporated in different domains.
> >
> > Additionally, we will provide metrics such as proximity, sparsity and validity that are often used to evaluate interpretability for counterfactual explanations, since this approach aligns more closely to ours that other interpretability methods.
> >
> >
> > ### ''Could the Tutor–Pupil setup be extended to adversarial tutoring, where the Tutor exposes brittleness or bias in the Pupil rather than assisting it?''
> >
> > This is a very interesting direction, and we have indeed considered such extensions. We chose not to include them in the current paper due to space constraints.
> >
> > We thank the reviewer again for these thoughtful questions!

---

### Official Review · Reviewer_SNr4 · 2025-11-05

**Soundness:** 3
**Presentation:** 4
**Contribution:** 2
**Rating:** 6
**Confidence:** 4

**Summary:**

The paper suggests a new form of interpretability approach without compromising expressivity. It is a common practice to use a simple interpretable model as a primary model and use a second model for handling the residual error for performance reasons. But interpretability is usually lost in the complexity of the second model.
This paper proposes an alternate. They use the second model instead to generate input perturbations such that the primary model is more accurate on the perturbed input. Assuming input perturbations are interpretable, the paper argues that their approach improves performance without hurting interpretability.

**Strengths:**

* Clean presentation. I enjoyed reading the paper.
* Neat conceptual difference. Modeling input perturbation instead of residual output error is a clean conceptual differential from earlier work.
* In the case of MNIST modeled with logistic regression, improving accuracy through augmentation and output correction could not have led to the explanations the paper demonstrated in Figure 6. MNIST dataset, although simple, supports their claim of improved accuracy and interpretability with their approach.

**Weaknesses:**

* Empirical validation. The validation in the paper looks preliminary. It requires validation with far more complex datasets to be taken seriously. For instance, CheXpert [1] or some WILDS [2] datasets.
* The paper assumes input edits are model-able and interpretable. The interpretability aspect is only assumed without validation.

References
[1] https://www.nature.com/articles/s42256-021-00338-7
[2] https://wilds.stanford.edu/datasets/

**Questions:**

1. If the pupil model is complex, what's stopping the tutor from selecting meaningless perturbations? I.e., how is the interpretability of perturbations enforced?
2. The requirement of latent space and pupil model inference with modified inputs could compromise performance due to train-test distribution for pupil model and other reasons. When do you expect far worser performance (than a non-interpretable base model) with your approach?
3. Please elaborate your differences from counterfactual explanations.

---

> ### Author Response · Authors · 2025-11-25
> **authors first resonse**
>
> We thank the reviewer for the time and effort they put in our article.
> We were particularly encouraged by their positive feedback on the clarity of our presentation and their appreciation of the MNIST example.
> The reviewer also highlighted several important points and questions, which we address below.
>
> ### ``If the pupil model is complex, what's stopping the tutor from selecting meaningless perturbations? I.e., how is the interpretability of perturbations enforced?''
>
>
> We agree with the reviewer that this is an important point.
> Indeed, it is natural to ask how we prevent the Tutor from producing perturbations that are technically effective but semantically meaningless.
>
> Our framework addresses this in two complementary ways:
>
>
> 1. Interpretability is encouraged through the *minimality* of perturbations.
> A central principle of our method is that the Tutor is trained to modify the input as little as possible while improving the Pupil's prediction.
> If minimal scattered perturbations are sufficient to alter the Pupil's decision, for example, barely noticeable pixel changes in an image, this still reveals something meaningful: the Pupil may suffer from robustness issues or may rely on spurious features. Thus, minimality may act both as a regularizer and as an interpretability constraint.
>
> 2. Interpretability can be enforced through *structured perturbation spaces depending on the domain.*
> Since the Tutor-Pupil framework is quite general, different applications allow different ways of constraining the Tutor's actions, as we tried to showcase in some of our examples:
>
> - In the ideal gas law example, the symbolic-regression to obtain a more general model step was applied without using any human interpretation of the corrections.
> We deliberately selected an example where the perturbations were inherently interpretable so that we could verify that the Tutor-Pupil framework behaved sensibly. The resulting data-driven analytic correction matched van der Waals-type terms, providing reassurance that the framework uncovers meaningful structure even without injecting the additional physical insights that historically lead to the formulation of van der Waals equation.
> - In the MNIST example, we operate in the latent space of a pretrained VAE. This low-dimensional, smooth manifold might naturally preserve semantic structure, helping the decoded perturbations remain visually interpretable.
> - In other applications, interpretability could instead be enforced through pixel segmentation, superpixels, or other human-aligned feature decompositions [Ribeiro 2016]. The framework is flexible enough to incorporate these constraints.
>
>
> In summary, the Tutor is not free to apply arbitrary corrections: interpretability arises from (i) enforcing minimal perturbations and (ii) constraining the perturbation space in domain-appropriate ways.
> We agree that this point deserves clearer emphasis. In the revised manuscript, we will highlight this aspect explicitly by including a discussion of how interpretability constraints can be incorporated in different domains.
>
>
> ### "Question: The requirement of latent space and pupil model inference with modified inputs could compromise performance due to train-test distribution for pupil model and other reasons. When do you expect far worser performance (than a non-interpretable base model) with your approach?"
>
>
> Thank you for the question. We are not entirely sure we fully understand its intent, but we address it as best we can here; if the Reviewer can clarify, we would be happy to respond further.
>
> In the Tutor-Pupil architecture, the key mechanism preventing degraded performance is the minimality of the Tutor’s corrections. This constraint can be controlled by hyperparameters that explicitly tune the trade-off between interpretability and accuracy.
>
> If this penalty is set very large, the Tutor is forced to produce vanishing perturbations ($\epsilon \approx 0$), and performance reduces to that of the Pupil alone, which can naturally be worse than that of a highly complex, non-interpretable model. Conversely, if the penalty is relaxed, the Tutor may introduce larger corrections and approach the performance of such complex models.
>
> Thus, the framework inherently spans the continuum between interpretability-oriented and accuracy-oriented regimes.

---

> > ### Author Response · Authors · 2025-11-25
> > **authors' first response**
> >
> > ### "Question: Please elaborate your differences from counterfactual explanations. "
> >
> > We thank the reviewer for this question.
> > Our framework is a model augmentation approach: the Tutor learns an input-space modification
> > $g(x)$ that becomes part of the predictive pipeline and is optimized to improve the Pupil's performance.
> >
> > Counterfactual methods, in contrast, are designed to explain a decision and do not aim to improve predictive accuracy.
> > For each instance $x$, they search for the closest $x'$ that changes the model's decision, and this search is performed independently for every data point of interest.
> >
> > The Tutor instead learns a global mapping
> > $g$ over the entire dataset by optimizing a loss that trades off performance improvement and the size of the modification. Since
> > $g(x)$ is learned jointly rather than independently for each instance, it provides a more holistic view of which input adjustments are broadly useful.
> >
> > We thank the reviewer again for raising this question. We will discuss this in the paper.
> >
> >
> > ### ''Empirical validation. The validation in the paper looks preliminary. It requires validation with far more complex datasets to be taken seriously. For instance, CheXpert [1] or some WILDS [2] datasets.''
> >
> > We thank the reviewer for providing examples that can be used to showcase our framework since large-scale examples are definitely an avenue of application for this framework that we'd like explore.
> > We just want to emphasize that the examples presented in the paper were deliberately kept simple to ensure that the underlying mechanisms of the Tutor–Pupil framework were transparent.
> > More complex datasets would inevitably introduce additional confounding influences and would require more specialized expertise to be understood.
> > For example the use of medical data would have made it difficult to validate the approach without expert knowledge and our goal was to keep focus on the augmentation approach.

---

### Meta-Review · Area_Chair_tS1u · 2026-01-08

**Summary:**

Strengths: Apparently effective interpretable scheme for augmenting models with a peripheral tutor-pupil system. Novel theory and promising experimental results.

Weaknesses:
- Interpretability is claimed but not evaluated against strong baselines.
- Only fairly small limited toy case studies are provided.
- MNIST performance boost could be due to the VAE latent representation and not the tutor’s corrections.
- 4yEq concerns about lack of novelty.

**Reviewer Concerns:**

- Authors adequately addressed questions of novelty.
- New experiments controlled the potential confounded for MNIST and supported the original results.


Outstanding:
- Authors argue that simple toy examples were good for illustrating their method. Nonetheless, it is unclear that the method would work or be better than other approaches in more realistic or large scale scenarios.
- While the authors do claim interpretability, their argument is that this method also improves model performance. I am somewhat sympathetic since they do show some improvements on evaluations, but I would like to see evidence that this is truly more interpretable than other performant options.

**Reviewer Scores:**

A couple reviewers apparently increased their scores. I do not think the score of 2 would have been raised, or it would have been raised minimally at best.

---

### Decision · Program_Chairs · 2026-01-26

Accept (Poster)